# The Molecular Mechanism and Therapeutic Application of Autophagy for Urological Disease

**DOI:** 10.3390/ijms241914887

**Published:** 2023-10-04

**Authors:** Kuang-Shun Chueh, Jian-He Lu, Tai-Jui Juan, Shu-Mien Chuang, Yung-Shun Juan

**Affiliations:** 1Graduate Institute of Clinical Medicine, College of Medicine, Kaohsiung Medical University, No. 100, Shih-Chuan 1st Road, San-min District, Kaohsiung 80708, Taiwan; ting.wei0220@gmail.com; 2Department of Urology, Kaohsiung Municipal Ta-Tung Hospital, Kaohsiung 80145, Taiwan; 3Department of Urology, College of Medicine, Kaohsiung Medical University, Kaohsiung 80708, Taiwan; u9181002@gmail.com; 4Center for Agricultural, Forestry, Fishery, Livestock and Aquaculture Carbon Emission Inventory and Emerging Compounds (CAFEC), General Research Service Center, National Pingtung University of Science and Technology, Pingtung 91201, Taiwan; toddherpuma@mail.npust.edu.tw; 5Kaohsiung Veterans General Hospital, Kaohsiung 81362, Taiwan; terry870921@gmail.com; 6Kaohsiung Armed Forces General Hospital, Kaohsiung 80284, Taiwan; 7Department of Urology, Kaohsiung Medical University Hospital, Kaohsiung 80708, Taiwan

**Keywords:** autophagy, urological disease, pathophysiological processes

## Abstract

Autophagy is a lysosomal degradation process known as autophagic flux, involving the engulfment of damaged proteins and organelles by double-membrane autophagosomes. It comprises microautophagy, chaperone-mediated autophagy (CMA), and macroautophagy. Macroautophagy consists of three stages: induction, autophagosome formation, and autolysosome formation. Atg8-family proteins are valuable for tracking autophagic structures and have been widely utilized for monitoring autophagy. The conversion of LC3 to its lipidated form, LC3-II, served as an indicator of autophagy. Autophagy is implicated in human pathophysiology, such as neurodegeneration, cancer, and immune disorders. Moreover, autophagy impacts urological diseases, such as interstitial cystitis /bladder pain syndrome (IC/BPS), ketamine-induced ulcerative cystitis (KIC), chemotherapy-induced cystitis (CIC), radiation cystitis (RC), erectile dysfunction (ED), bladder outlet obstruction (BOO), prostate cancer, bladder cancer, renal cancer, testicular cancer, and penile cancer. Autophagy plays a dual role in the management of urologic diseases, and the identification of potential biomarkers associated with autophagy is a crucial step towards a deeper understanding of its role in these diseases. Methods for monitoring autophagy include TEM, Western blot, immunofluorescence, flow cytometry, and genetic tools. Autophagosome and autolysosome structures are discerned via TEM. Western blot, immunofluorescence, northern blot, and RT-PCR assess protein/mRNA levels. Luciferase assay tracks flux; GFP-LC3 transgenic mice aid study. Knockdown methods (miRNA and RNAi) offer insights. This article extensively examines autophagy’s molecular mechanism, pharmacological regulation, and therapeutic application involvement in urological diseases.

## 1. Introduction

Autophagy is a highly regulated cellular process that is crucial for maintaining cell homeostasis by clearing damaged organelles and proteins [1]. In recent years, research efforts have increasingly focused on unraveling the molecular mechanisms of autophagy and its potential therapeutic applications in various urological diseases, such as interstitial cystitis/bladder pain syndrome (IC/BPS), ketamine ulcerative cystitis (KIC), chemotherapy-induced cystitis (CIC), radiation cystitis (RC), erectile dysfunction (ED), bladder outlet obstruction (BOO), prostate cancer, bladder cancer, renal cancer, testicular cancer, and penile cancer. The role of autophagy in these urologic diseases is complex and multifaceted, offering both protective and detrimental effects, depending on the specific context. This article aims to provide a comprehensive overview of the molecular mechanisms underpinning autophagy and explore its potential as a therapeutic strategy in managing urological diseases. 

## 2. Overview of Autophagy

### 2.1. Comparison between Autophagy and Apoptosis

Apoptosis is genetically programmed cell death and occurs under both physiological and pathological conditions, including cell injury, stress, development, immune response, and cancer treatment. It encompasses both intrinsic and extrinsic pathways. The intrinsic pathway, distinguished by the liberation of cytochrome c from the mitochondria, plays a crucial role in triggering caspase activation. Cytochrome c is released and bound to the caspase-activating protein Apaf-1 to form a complex called the apoptosome. In the extrinsic pathway, the activation of cell surface death receptors, such as Fas, occurs upon binding to their specific ligands [2]. This receptor–ligand interaction initiates caspase-8 activation, which subsequently causes a cascade of intracellular events, resulting in cell death [3]. Mitoptosis representes an apoptotic-like phenomenon occurring within mitochondria, leading to mitochondrial outer membrane permeabilization (MOMP) and the possibility of mitochondrial loss. 

Autophagy assumes a vital function in upholding cellular homeostasis and promoting cell survival [1]. This is achieved through the degradation and repurposing of compromised cytoplasmic elements via lysosomal mechanisms, encompassing impaired and harmful aggregated proteins, as well as dysfunctional organelles, such as mitochondria and the endoplasmic reticulum (ER). It eliminates damaged organelles, such as mitochondria, which have the potential to impede the generation of ROS capable of harming cellular DNA [4]. Mitophagy refers to the mechanism by which abnormal mitochondria are identified and eliminated through autophagy-regulated degradation. Recent investigations have indicated the importance of mitochondrial fission, a process controlled by the GTPase dynamin-related protein 1 (Drp1), in the regulation of mitophagy. Various stress stimuli, such as TNF-α [5], reactive oxygen species (ROS) [6], ER stress [7,8], DNA damage, misfolded protein aggregation, mitochondrial dysfunction [9,10], nutrient starvation [11], ischemia [12], inflammation, pathogen infection, and metabolic stress [13], activate the autophagic signaling pathway. Clinically, autophagy has emerged as a factor implicated in a diverse range of human pathological diseases, including cancer [14,15,16,17], neurodegenerative disease [17,18,19,20,21], acute kidney injury [22,23], metabolic disease [13,24], inflammatory disease [24,25,26,27], heart disease [28], myopathy [17,29], autoimmune disease, and pathogen infection [17,30]. Whether autophagy assumes a protective or detrimental role in human diseases remains an aspect that lacks clear establishment.

Both apoptosis and autophagy are triggered in reaction to metabolic stress, including growth factor, nutrient and energy deprivation, ER stress, and activation of the LKB1–AMPK pathway [31]. Interestingly, disturbing the balance of ER calcium levels or impairing ER functionality can lead to heightened autophagy and apoptosis. A comparison between autophagy and apoptosis is shown in Table 1. The influence of autophagy on cellular survival under ER stress varies based on tissue type. For instance, in colon and prostate cancer cells, ER-triggered autophagy serves a critical function in the removal of undesirable polyubiquitinated protein aggregates, thereby safeguarding against cell demise. Moreover, squamous cancer cells experience a dual occurrence of apoptotic and autophagic cell death, both of which are regulated by the LKB1/AMPK pathway [31]. Accumulated data suggest that autophagy and apoptosis can interact in various ways, cooperating, antagonizing, or mutually supporting each other, thereby differentially influencing the ultimate fate of the cell. Furthermore, autophagy can exert an impact on apoptotic cascades by regulating caspases. Diverse pro-apoptotic pathways have the capability to activate caspases, initiating the apoptotic process. In addition, once activated, caspases have the capacity to cleave and degrade essential autophagic proteins, including Beclin-1, p62, Atg3, Atg4D, Atg5, Atg7, and AMBRA1. Certain molecules play dual roles in both apoptosis and autophagy. Examples include Atg5 and Beclin-1, which interact with Bcl-2/Bcl-XL and participate in type III PI3-kinase-mediated autophagosome membrane formation, essential for autophagosomal membrane generation and recruitment of Atg8. In pathological contexts, the interplay between autophagy and apoptosis can be pertinent to certain diseases. Notably, in colon and prostate cancer cells, autophagy induced by ER stress plays a crucial role in eliminating unwanted polyubiquitinated protein aggregates, thereby shielding against cell demise. Additionally, squamous cancer cells experience a combination of apoptotic and autophagic cell death, orchestrated by the regulatory influence of LKB1/AMPK [31]. Evidence suggests a connection between apoptosis and autophagy through shared regulatory elements. The identification and characterization of molecules and pathways that bridge these processes hold promise for unveiling novel therapeutic avenues in addressing cancer and neurodegenerative disorders. Autophagy precedes apoptosis as a defense mechanism to reestablish homeostasis [17].

Unlike autophagy and apoptosis, necrosis represents a form of cell demise induced by injury, resulting in the rupture of cells and the release of their intracellular contents, consequently harming adjacent cells. This process triggers inflammation within nearby tissues, compounding the overall damage and trauma. Necrosis is an unregulated, accidental form of cell death. It can be stimulated by various factors, such as trauma, ischemia, infection, exposure to toxins, thermal or chemical burns, and certain autoimmune reactions. Based on morphological changes, necrosis can be classified into several types, including coagulative, liquefactive, caseous, fat necrosis, fibrinoid necrosis, and gangrenous necrosis [32]. The morphological characteristics of necrosis encompass heightened eosinophilia, a reduction in cytoplasmic RNA content, and an elevated affinity of eosin staining to denature cytoplasmic proteins. Each type of necrotic characteristic is associated with specific pathological conditions.
ijms-24-14887-t001_Table 1Table 1Comparison between autophagy and apoptosis.CharacteristicsAutophagyApoptosisDefinitionA catabolic process that entails the breakdown of cellular components utilizing the lysosomal machinery.A genetically orchestrated form of programmed cell death, including the initiation of proteases (such as caspases) and nucleases within an intact plasma membrane.Morphological hallmarksNutrient starvation.Double membrane autophagic vacuoles.Lysosomal protein degradation.Cellular stress.Energy dependent.No inflammatory response.Cell shrinkage.Chromatin condensation.Chromosomal DNA fragmentation.Membrane blebbling (apoptotic bodies).No inflammatory response.Energy dependent.Mechanism or subtypeMacroautophagy (major subtype): The process entails the enclosure of cytplasmic components within double -membrane autophagosomes, which subsequently merge with lysosomes for degradation and recycling.Microautophagy.Chaperone-mediated autophagy.Intrinsic pathway: Signaling is induced by hypoxia, DNA damage, oxidative stress, and growth factor deprivation.Extrinsic pathway: Signaling is initiated by activating the death receptors.Major playersAutophagy related genes, e.g., *Beclin 1*.Class I and III PI3K pathway.TOR-negative regulator.Caspases.ER.Mitochondrial autophagy (mitophagy): programmed destruction of mitochondria.P53.Intrinsic pathway (Cytochrome c, Apaf1, caspase-9).Extrinsic pathway (death receptors, FADD, caspase-8/10).TNFα.Caspases.ER.Mitochondrial apoptosis (mitoptosis).P53.Human diseaseNeurodegenerative diseases [18,19,20], cancer [14,15], inflammation, autoimmune disease [33], acute kidney injury [22,23], myopathies, heart diseases, septic liver diseases [34], and pathogen infection [17,35].Neurodegenerative diseases [18,19,20], cancer [14], heart diseases, autoimmune disease [33], bacterial and viral diseases [28,35,36]


### 2.2. Molecular Mechanisms of Autophagy

There are primarily three types of autophagy: (1) microautophagy, which directly sequesters and engulfs the cytoplasmic components by inward invagination of the lysosome membrane [37]; (2) chaperone-mediated autophagy (CMA), in which cytosolic proteins containing the KFERQ-like motif are recognized by chaperones, unfolded, and translocated into the lysosome [38]; and (3) macroautophagy, identified by the creation of autophagosomes (compartments enclosed by a double membrane), which subsequently merge with lysosomes to transport the contents from the cytoplasm [39]. Microautophagy involves the direct transfer of cytosolic components into the lysosome, followed by the budding of vesicles into the lysosomal lumen. CMA is characterized by a selective degradation system regarding the specific substrates (cytosolic proteins) degraded [40]. Cytosolic proteins containing KFERQ-like motifs are recognized by the cytosolic chaperone Hsc73. Upon binding to the lysosomal receptor Lamp2a, the resulting complexes cause the unfolding and translocation of substrates into the lysosomal lumen, facilitating their degradation [41]. Macroautophagy participates in the formation of a double-membrane structure known as the autophagosome. This structure encloses cytosolic components, such as misfolded protein aggregates and organelles, subsequently transporting them to the lysosome for degradation. Although the degradation for specific removal of damaged mitochondria might be selective, the degradation for soluble cytosolic proteins might be non-selective.

Autophagy is a dynamic process referred to as autophagic flux. The process of macroautophagy has three distinct stages as follows: autophagy induction, autophagosome (phagophore) formation, and autolysosome formation [42,43] (Figure 1). (1) Autophagy induction (initiation): Autophagy induction involves the creation and expansion of the phagophore, the initial sequestering compartment that eventually grows into an autophagosome. During periods of stress or nutrient deprivation, the inhibition of the mammalian target of rapamycin (mTOR) triggers autophagy. mTOR, a Ser/Thr signaling kinase, plays a pivotal role in the regulation of autophagosome formation. The process is achieved by the assembly of the Unc-51-Like Kinase (ULK) complex, consisting of ULK, Atg13, and FAK-family interacting protein of 200 kDa (FIP200). Subsequently, the ULK complex phosphorylates the activating molecule AMBRA1 in the context of Beclin-1-regulated autophagy. This, in turn, triggers the activation of the phosphatidylinositol-3-kinase (PI3K) complex, comprising Beclin-1, VPS15, VPS34, and AMBRA1 [44,45]. During autophagy induction, VPS34 and Beclin-1, known as Atg6, are phosphorylated. (2) Autophagosome formation: The extension and enlargement of the phagophore membrane are governed by two systems of conjugation similar to ubiquitin: Atg12-Atg5-Atg16 and Atg8-Atg3. In the Atg12-Atg5-Atg16 system, the process begins with the activation of Atg12 by the E1-like activating enzyme Atg7, which relies on ATP. Subsequent interaction between the Atg5-Atg12 heterodimer and the Atg16 homodimer culminates in the formation of the Atg12-Atg5-Atg16 complex. The second ubiquitin-like system facilitates the binding of phosphatidylethanolamine (PE) to Atg8/microtubule-associated protein 1 light chain 3 (LC3). This modification is then processed sequentially by Atg4, Atg7, and Atg3. In the context of autophagy, the cytosolic LC3 form (nonlipidated LC3-I) becomes linked to PE, resulting in the lipidated LC3 (LC3-II) form, which is aggregated on the autophagosomal membrane as an autophagy marker [28]. (3) Autolysosome formation: The genesis of autolysosome formation arises from the transportation and subsequent fusion of the autophagosome with the lysosome. The autophagosome membrane elongates and matures, and LC3 is recruited to the membrane. The double-membrane autophagosomes forms and sequesters cytoplasmic proteins and organelles, such as mitochondria. TEM observations have revealed that the autophagosome exhibits a dual-membrane structure enclosing undigested cytoplasmic elements, including organelles. In contrast, the autolysosome is characterized by a solitary membrane structure encompassing cytoplasmic components. The autophagosome merges with a lysosome, which houses acid hydrolases (AH). The fusion process between the autophagosome and lysosome necessitates the presence of acetylated microtubules [46]. The resultant merged compartment, where the autophagic body and cytoplasmic components undergo degradation, is termed an autophagolysosome or autolysosome. After degradation, the macromolecules are liberated into the cytosol via permeases, allowing for their recycling during metabolic stress and nutrient deprivation. 

### 2.3. Autophagy Related Gene (Atg)

A family of autophagy related genes /proteins were originally identified in yeast and are crucial for autophagy regulation. Mammalian homologues of autophagy-related (Atg) proteins have been identified (Table 2). At least 13 *Atg* genes are required for autophagosome formation [28,47,48].

### 2.4. Autophagy Biomarkers

Biomarkers play an important role in the diagnosis, prognosis, and treatment of urological diseases [59,60]. Autophagy biomarkers serve as crucial indicators of autophagy in various contexts, including normal physiological processes, pathological conditions, and responses to pharmacological treatments. Among these markers, Atg8-family proteins are particularly valuable for tracking autophagic structures and have been widely utilized for monitoring autophagy [61]. The conversion of LC3-I to LC3-II through conjugation with phosphatidylethanolamine serves as a widely adopted indicator of autophagy [62]. LC3B serves as the mammalian homologue to yeast Atg8. Beclin1 and Atg8/LC3B are typical autophagy biomarkers that are frequently employed to gauge the extent of autophagic activity [63,64]. In addition, Atg5, Atg12, and Atg16L1 are associated with the phagophore [65]. Therefore, endogenous Atg proteins form puncta and monitor autophagy upregulation. 

### 2.5. Pharmacological Manipulation for Autophagy 

Using pharmacological manipulation (Figure 1, and Table 3), we can enhance or inhibit autophagic expression. Several interventions of autophagy have the potential to elevate the number of autophagosomes: (1) Disrupting the process that activates MTORC1 through lysosomes, which is a key suppressor of autophagy initiation [66,67,68]; (2) Inhibiting lysosome-mediated proteolysis, achieved through measures such as employing the cysteine, serine, and threonine protease inhibitor leupeptin or utilizing agents, such as bafilomycin A1, NH4Cl, or Chloroquine [69,70] (Figure 1); (3) Hindering the fusion of autophagosomes with lysosomes, for instance, through bafilomycin A1 treatment [71].

Rapamycin functions as an mTOR inhibitor, stimulating autophagy by hindering the phosphorylation of Akt. It accomplishes this by binding to mTOR, thereby impeding the phosphorylation of downstream targets, such as p70 ribosomal protein S6 kinase (P70S6K) and eukaryotic translation initiation factor 4E-binding protein 1 (4E-BP1). These downstream substrates are implicated in processes such as transcription, translation, and cell cycle control, and subsequently influence cell survival or apoptosis [72]. Rapamycin is also a potent immunosuppressive and anti-proliferative property [73]. Moreover, tamoxifen increases Beclin 1 and induces autophagy (Figure 1, Table 2). 

On the other hand, 3-methyladenine (3-MA) and wortmannin act as inhibitors of Class III PI3K to restrain the formation of the pre-autophagosomal structure. These inhibitors exert their autophagy-suppressing effects by impeding Class III PI3K activity, thus obstructing the generation of phosphatidylinositol 3-phosphate (PI3P), which holds significance in the initiation of autophagy [74]. The process of autophagic sequestration necessitates ATP and is under the regulation of Class III PI3K. Bafilomycin A1, acting as a vacuolar H^+^-ATPase inhibitor, serves to obstruct autophagy. Beclin 1 (Atg6) is a constituent of the PI3K complex, whose activation spurs the initiation of autophagosomal membrane nucleation. Autophagosomes progress to maturity through acidification enabled by the H^+^-ATPase and subsequently merge with lysosomes, leading to the formation of autolysosomes or degradative autophagic vacuoles. Notably, an intact microtubule network contributes to the maximal formation and movement of autophagosomes, as well as their fusion with endosomes [46,75]. The autophagic sequestration process is hindered by the H^+^-ATPase inhibitor bafilomycin A1, as well as by microtubule inhibitors, such as vinblastine and nocodazole (Figure 1, Table 3). Bafilomycin A1 induces both a block in the fusion and neutralization of the pH within lysosomes [76]. However, the inhibition of fusion may be derived from a block in ATP2A/SERCA activity [77]. Chloroquine inhibits autophagic flux by reducing the fusion between autophagosomes and lysosomes [78]. Yamamoto and colleagues, in their study involving the rat hepatoma cell line H-4-II-E cells, showed that bafilomycin A1 had the effect of hindering the maturation of autophagic vacuoles. This inhibition was achieved by interfering with the fusion process between the autophagosomes and lysosomes.
ijms-24-14887-t003_Table 3Table 3Pharmacological regulation of autophagy activators and inhibitors.CompoundTarget and EffectReferenceAutophagy activatorsRapamycinAn inhibitor of the Ser/Thr protein kinase, mTOR inhibitor.Induce autophagy.[79,80,81,82]TamoxifenIncrease Beclin 1.Induce autophagy.[83]Brefeldin AThapsigarginTunicamycinER stressing inducer.Autophagy induction.[84,85,86,87]Autophagy inhibitorsWortmanninLY294002Class III PI3K inhibitor.Sequestration of autophagy is regulated mainly by class III PI3K.Inhibit autophagy (autophagosome formation).[74,88,89,90]3-methyladenine(3-MA)A Class III PI3K inhibitor.3-MA blocks the initiation stage of autophagy by inhibiting class III PI3K.Inhibit autophagy (autophagosome formation).3-MA does not inhibit BECN1-independent autophagy.[74,91,92,93,94]Bafilomycin A_1_Vacuolar H^+^-ATPase inhibitor; An inhibitor of autophagosomelysosome fusion.Bafilomycin A_1_ causes elevated pH within lysosomes/vacuoles and obstructs the fusion of autophagosomes with the vacuole.Inhibit autophagy.[95,96]Vinblastine NocodazoleMicrotubule inhibitors to inhibit autophagy.Microtubules play a crucial role in facilitating this fusion process.[46,75]Carbamazepine LithiumSodium valproateTriggers autophagy by suppressing inositol monophosphatase. [97,98]ChloroquineHydroxychloroquineLysosomal lumen alkalizer: Inhibited lysosomal acidification and elevate/neutralize the lysosomal/vacuolar pH.They hinder the fusion between autophagosomes and lysosomes, as well as the degradative activity within lysosomes.Inhibit autophagy.Chloroquine and hydroxychloroquine are being evaluated in clinical trials to determine their therapeutic efficacy when used in conjunction with cancer chemotherapy.[78,99] CycloheximideA protein synthesis inhibitor.Inhibit autophagy (autophagosome formation).[100,101]LeupeptinE64dPepstatin AAn acid protease inhibitor.Suppressed lysosome to inhibit autophagolysosome formation.[102,103]Note: Atg, autophagy-related gene; ULK, Unc-51–like kinase 1; PI3K, class III phosphotidylinositol- 3-kinase; PI3P, phosphatidylinositol-3-phosphate.


## 3. Potential Therapeutic Application of Autophagy in Urological Diseases

Autophagy has been linked to various urologic conditions, and the precise role it plays in these pathologies remains incompletely understood. In this article, we showed the potential therapeutic strategy of autophagy for urological diseases in human and animal models, including interstitial cystitis/bladder pain syndrome (IC/BPS), ketamine-induced ulcerative cystitis (KIC), chemotherapy-induced cystitis (CIC), radiation cystitis (RC), erectile dysfunction (ED), bladder outlet obstruction (BOO), prostate cancer, bladder cancer, renal cell carcinoma, testicular cancer, and penile cancerer.

### 3.1. Cystitis

Cystitis can be classified into two main types: bacterial cystitis and non-bacterial cystitis. Non-bacterial cystitis encompasses several subtypes, including IC/BPS, CIC, KIC, and RC [104]. In experimental animal models, CIC and KIC are both drug-induced conditions utilized to mimic the symptoms and characteristics of IC/BPS. CIC and KIC share similar symptoms, including suprapubic pain during bladder distention, urinary frequency, urinary urgency, and gross hematuria. Treating CIC and KIC remains a challenging issue, including lifestyle modification, physical therapy, medication, and bladder instillation with hyaluronic acid.

#### 3.1.1. Interstitial Cystitis/Bladder Pain Syndrome (IC/BPS)

IC/BPS is diagnosed as symptoms persisting for at least 6 weeks, including suprapubic pain and/or discomfort during bladder distention, urinary frequency, and urinary urgency, with or without urge incontinence [105,106]. Research findings indicate that persistent inflammatory stimulation can trigger epithelial to mesenchymal transition and promote a pro-fibrogenesis phenotype in IC/BPS [107]. Additionally, certain studies have indicated that autoimmune responses targeting urothelial antigens may play a role in bladder inflammation and the pathology of IC/BPS [108]. Bladder tissue obtained from patients with IC displays characteristics such as a thinner and swollen epithelium, along with the presence of inflammatory infiltration in the lamina propria and muscle layer [109,110,111]. Abundant mast cell infiltration in the muscle layer has also been observed [111]. Zhao et al. observed a significant increase in autophagy within detrusor myocytes in IC/BPS patients. Moreover, in the urothelium of IC/BPS subjects, the upregulation of pro-apoptotic molecule phospho-p53 (Ser 15), Bad, Bax, cleaved caspase-3, Fas, and cleaved caspase-8 is shown [112,113]. Additionally, in individuals with IC, the expression of the LC3 protein, recognized as a marker of autophagy, exhibits a notable increase in bladder tissue. This augmentation has been confirmed through techniques such as double-labeled immunofluorescence and Western blot analysis [114].

#### 3.1.2. Chemotherapy-Induced Cystitis (CIC)

Cyclophosphamide (CYP) is a potent chemotherapy medication used to treat certain autoimmune disorders and cancer, such as leukemia and lymphoma [115]. Ni et al. demonstrated that CYP-treated rats increased urinary frequency and urgency, pain sensitization, decreased bladder contractility, bladder edema, and oxidative stress. However, the rats showed significantly improved bladder micturition function and oxidative stress after treatment with rapamycin [116]. In addition, autophagosomes have been shown in detrusor myocytes, and the expression of LC3 and p-p70s6k are increased in CYP-induced cystitis rats. In a prior study, the CIC rats exhibited reduced autophagic activity within detrusor smooth muscle, increased infiltration of mast cells, and elevated expression of inflammatory factors (IL-6, IL-8, IL-1β), contributing to impaired urinary function [116]. Following rapamycin administration, an activator of autophagy, there was a significant reduction in the infiltration of mast cells of the bladder and the expression of inflammation markers (IL-6, IL-8, IL-1β) in bladder [116]. The study revealed that autophagy played a protective role in bladder function by inhibiting inflammation in the detrusor muscle layer.

#### 3.1.3. Ketamine-Induced Ulcerative Cystitis (KIC)

Ketamine administration results in frequent bladder contraction and can be used as a potential rat model for an overactive bladder (OAB). Clinical investigations have indicated that individuals who use ketamine may experience pronounced lower urinary tract symptoms (LUTS), including heightened urinary frequency, nocturia, urgency, bladder pain, dysuria, and occasionally hematuria. These symptoms bear a striking resemblance to those associated with IC/BPS [117,118,119]. Ketamine-addiction patients exhibit distinctive pathological features during endoscopy, including observations of bladder erythematous mucosa, mucosal ulceration and laceration, wall thickening, hydroureter, and hydronephrosis, as well as swelling of the ureter mucosa [120,121]. In clinical cases of KIC, the patients display heightened infiltration of mast cells and eosinophil cells within the bladder. Additionally, there is an elevation in serum immunoglobulin-E (IgE) levels, which is associated with hypersensitivity and/or allergic reactions [122,123,124].

In the KIC rat model, ketamine treatment significantly causes bladder overactivity, aggravates interstitial fibrosis, impairs angiogenesis, triggers eosinophil-mediated inflammation, swelling, and mitochondrial degradation, and elevates the phosphorylation of Akt [125,126,127,128]. Bladder tissues exhibit heightened expression levels of Atg proteins, including Atg12, Atg7, Beclin1, LC3, and VPS34. These levels are significantly increased in the ketamine + rapamycin group compared to the ketamine-only group. Furthermore, treatment with rapamycin ameliorated eosinophil-mediated inflammation, thereby improving the pathogenesis of KIC. On the other hand, treatment with wortmannin reduced basophil-mediated inflammatory responses [127]. The administration of rapamycin exerts an inhibitory impact on vascular formation, facilitates the elimination of ketamine metabolites, reduces eosinophil-mediated inflammation, and alleviates bladder hyperactivity. Consequently, it contributes to the enhancement of bladder function in KIC. Conversely, wortmannin enhances bladder angiogenesis by elevating capillary density and upregulating VEGF expression. This counteracts the anti-angiogenic effect, promoting the repair of KIC [127]. We suggest that the dual effect of autophagy on anti-angiogenesis is contingent upon factors such as different cell types, cellular requirements, and prevailing conditions.

#### 3.1.4. Radiation Cystitis (RC)

RC is caused by radiotherapy used to treat pelvic cancers, such as prostate and bladder cancer. Radiation damage in the bladder leads to inflammation and injury [129]. In RC, the symptoms may encompass urinary frequency, urgency, pain, or discomfort during urination, gross hematuria, and bladder spasms [129]. Clinical management of storage symptoms in RC primarily involves symptomatic approaches using analgesics and anti-inflammatory drugs to alleviate discomfort. Treatment for RC often includes medications for symptom relief, adjustments in lifestyle, and specialized interventions, such as hyperbaric oxygen therapy. Preventing urinary obstruction due to blood clots is also a key consideration [130] or endoscopic procedures to control bleeding or alleviate bladder inflammation [131].

Clinical management of storage symptoms in RC primarily involves symptomatic approaches using analgesics. The level of autophagy has been found to increase significantly during radiotherapy, and appropriately induced autophagy can enhance the therapeutic effect. Hepatoma cells show an increase in the number of autophagosomes observed by electron microscopy after radiation. Wu et al. [132] investigated the interaction between radiation and autophagy in oral squamous cell carcinoma cell lines (OC3). The results indicate that the survival rate of OC3 cells decreased when co-treated with radiation and the autophagy-inducing agent rapamycin, implying a synergistic effect of radiation and autophagy induction on OC3 tumor cell growth. Furthermore, the pro-apoptotic impact of Saikosaponin-D on hepatoma cells induced by radiotherapy was notably reversed after treatment with chloroquine or an mTOR agonist. Another study by Wang et al. demonstrated that Saikosaponin-D enhanced radiation-induced apoptosis in hepatoma cells by augmenting autophagy through the inhibition of mTOR phosphorylation and anti-inflammatory drugs to alleviate discomfort. Treatment for RC often includes medications for symptom relief, adjustments in lifestyle, and specialized interventions, such as hyperbaric oxygen therapy [131].

### 3.2. Erectile Dysfunction (ED)

Clinically, ED is defined as the inability to achieve or maintain an erection of the penis that is sufficient for satisfactory sexual intercourse for the duration of at least 3 months [133,134]. ED is treatable by a phosphodiesterase type 5 inhibitor. The contraction/relaxation of corpus cavernosum smooth muscle cells (SMCs) plays a critical role in penile erection. Both autophagy and apoptosis are important processes for the pathogenesis of diabetic ED. Autophagy inducers and inhibitors can be utilized as pharmacological agents to modulate angiogenesis. In addition, the decline in autophagic activity associated with aging is connected to ED [135]. Panaxnotoginsengsaponin has a positive effect on ED by stimulating the autophagy pathway and upregulating the phosphorylation of gap junction protein, connexin 43. In corpus cavernosum SMCs of rats with ED, there is a decrease in the expression of Beclin-1 and an increase in the levels of apoptosis markers P62 and cleaved caspase-3 [136].

ED is prevalent in about 50% of men with diabetes mellitus (DM). Vascular complications associated with diabetes involve vascular endothelial damage. In rats with type 1 diabetes induced by streptozotocin, the administration of rapamycin has shown potential to improve erectile dysfunction. This improvement is attributed to the induction of autophagy and the inhibition of apoptosis, as well as the attenuation of endothelial dysfunction and corporal fibrosis [137]. The autophagic activity in the rat corpus cavernosum can be evaluated by examining the levels of LC3-II, mTOR, and p62. An increase in LC3-II levels and a decrease in mTOR activity, along with elevated p62 levels, collectively indicate the activation of autophagy in this tissue. This assessment helps provide insights into the autophagic process taking place in the corpus cavernosum, which can be relevant to conditions such as erectile dysfunction [138]. Rapamycin has demonstrated a positive impact on erectile function in rats with diabetic ED by enhancing autophagy, inhibiting apoptosis and fibrosis, and improving endothelial function. The AMPK/mTOR and PI3K/AKT/mTOR signaling pathways have been found to be activated in the diabetic ED group. However, compared to the diabetic ED group, the administration of rapamycin results in reduced expression of the AMPK/mTOR and AKT/mTOR pathways [137]. Additionally, in rat mesenchymal stem cells (MSCs), low-intensity shock wave therapy (LiESWT) has been shown to stimulate autophagy through the PI3K/AKT/mTOR pathway. When LiESWT is combined with MSC therapy, it leads to increased levels of VEGF compared to single treatments. This combined therapy further engages in autophagy by triggering the PI3K/AKT/mTOR signaling pathway. This innovative approach of combining LiESWT with MSC therapy holds potential as a novel avenue for the research and treatment of erectile dysfunction [139].

### 3.3. Bladder Outlet Obstruction (BOO)

Clinically, BOO, defined as a high-pressure and low-flow micturition pattern at urodynamic examination, is a common causative factor in humans with benign prostatic hyperplasia (BPH). BPH is the common urological etiology (46.8%) of BOO [140]. Clinical symptoms of BOO include increased voiding frequency and urgency, hesitancy, incomplete bladder emptying, and residual urine [141]. BOO results in changes in bladder structure and function, including elevated detrusor contractile pressure and muscle hypertrophy, which can lead to OAB [142,143]. BOO triggers a process of bladder wall remodeling, which is characterized by changes in bladder SMCs due to altered hydrostatic pressure (HP). This leads to hypertrophy of SMCs and an excessive accumulation of extracellular matrix (ECM) in the bladder. Interestingly, autophagy plays a role in regulating the biological function of bladder SMCs and contributes to mitigating the excessive deposition of ECM [144]. A growing body of evidence indicates that autophagy is intricately involved in the bladder remodeling that occurs as a result of BOO. This is due to its dual nature, which encompasses both pro-cell survival and pro-cell death properties [145].

A previous study demonstrated that the administration of rapamycin, an mTOR inhibitor, led to a reduction in bladder overcapacity, mitigated smooth muscle hypertrophy, and improved the pathological condition characterized by increased residual urine volumes in cases of long-term obstruction [146]. Inhibition of mTOR in vivo can potentially reduce the detrusor hypertrophy induced by obstruction and help maintain normal bladder function. Survivin, a protein that participates in both physiological and pathological processes, such as wound healing, neovascularization, and scar formation, has a dual impact by encouraging the cell cycle while obstructing apoptosis [147,148,149]. Survivin is a member of the inhibitor of apoptosis protein (IAP) family and has been linked to autophagy regulation. When survivin is knocked down, the decrease in autophagy levels induced by pathological HP is reversed, as evidenced by changes in the LC3B II/I ratio and Beclin1 expression. Furthermore, an increase in survivin expression is accompanied by the upregulation of autophagy-related factors, such as LC3, Beclin1, and p62. Inhibiting survivin can counteract the hyperglycemia-induced overactivity of endothelial cells. By utilizing autophagy inhibitors (such as 3-MA) and agonists (such as rapamycin), it has been demonstrated that survivin acts downstream of autophagy. This suggests that targeting the survivin/autophagy axis could hold the potential for therapeutic interventions in the treatment of diabetic vascular complications [150].

### 3.4. Urological Cancer

Monoallelic disruption of Beclin1 has been observed in a substantial proportion (ranging from 40% to 75%) of various human tumors, including those in the breast, ovaries, and prostate. Clinical studies have revealed that abnormal expression of Beclin 1 in tumor tissues is linked to poor prognosis for certain tumor phenotypes. Further research conducted by Yue and colleagues confirmed that mice with a heterozygous Beclin 1 mutation (Beclin 1^+/−^) exhibited a high incidence of tumor development [151,152]. FOXO1, a member of the forkhead O (FOXO) family of proteins, interacts with Atg7 to regulate autophagy. It plays a crucial role in initiating autophagy and suppressing tumorigenesis in human colon tumors [153]. In leukemic cells, when lysosomal-associated membrane protein-2 (LAMP2) is inhibited, there is a noticeable increase in GFP-LC3 puncta and endogenous LC3-II protein levels compared to control cells. This increase has been observed during autophagy induction as part of the myeloid differentiation process.

Inhibiting autophagy for cancer therapy includes surgery (growth factor limitation, nutrient limitation, and hypoxia), chemotherapy, targeted therapies, and radiation [154,155]. Autophagy can potentially aid cancer cells in resisting the harmful effects of ionizing radiation by eliminating damaged cellular components, such as mitochondria [156]. Recent research indicates that autophagy plays a significant protective role in cancer treatment and can influence the response of cancer cells to chemotherapy, potentially contributing to resistance. Reduced autophagy has been associated with tumor progression, while certain tumor-suppressor proteins regulate autophagy to restrain tumor growth (examples include Beclin-1 and PTEN). Moreover, the impact of autophagy on cancer migration and invasion is complex, as it can both promote and inhibit these processes across different types of malignancies [157,158]. For instance, rapamycin has been applied to Phase III clinical trials for glioma therapy [159,160], and used to treat hamartomas associated with tuberous sclerosis [161]. Similarly, tamoxifen has an anticancer effect by upregulating the level of Beclin 1 and inducing autophagy [83]. Autophagy plays a significant role in promoting cancer metastasis through its ability to induce the epithelial-to-mesenchymal transition. When autophagy is inhibited, the levels of mesenchymal markers, such as vimentin and CDH2/N-cadherin decrease, further confirming the influence of autophagy on cancer metastasis [162]. As a result, autophagy has been identified as a promising target in cancer treatment to improve both mortality and morbidity. However, it is worth noting that autophagy’s impact on cancer is complex and multifaceted. Although it can either enhance or suppress cancer progression, its exact role remains a topic of ongoing investigation and is not fully resolved [163].

#### 3.4.1. Prostate Cancer

Prostate cancer is the leading cause of urological malignancies worldwide seen in men. Autophagy can regulate the proliferation and metastasis capacity of prostate cancer cells [151,164]. The damage to Beclin1 (a tumor suppressor) is related to prostate cancer. ATG5 expression is upregulated in prostate carcinomas [165,166]. Autophagy induced by ER stress plays a critical role in prostate cancer cells by preventing apoptosis through the aggregation of polyubiquitinated proteins. When cancer cells are treated with chloroquine derivatives, lysosomes lose their acidity, resulting in the accumulation of autophagic vesicles. Manipulating autophagy can affect how prostate cancer cells respond to chemotherapy and radiotherapy, potentially influencing their therapeutic outcomes. Recently, the therapeutic application of autophagy in prostate cancer with promises for clinical application includes prognosis, diagnosis, chemotherapy, radiotherapy, immunotherapy, metastasis, and apoptosis activation [167].

Further investigations have elucidated the impact of androgen deprivation therapy on prostate cancer in relation to autophagy. The androgen receptor (AR) signaling pathway plays a significant role in the progression of prostate cancer by facilitating the process of autophagy [168,169]. Androgen deprivation or treatment with the AR-signaling inhibitor Enzalutamide induced autophagy in androgen-dependent and castration-resistant prostate cancer (CRPC) cell lines. The autophagic cascade triggered by AR blockage is correlated with the increased LC 3-II/I ratio and ATG-5 expression. Autophagy is an important mechanism of resistance to AR signaling inhibitors in CRPC. In addition, antiandrogen-induced autophagy is mediated through the activation of the AMPK pathway and the suppression of the mTOR pathway [170].

The downregulation of GABARAPL1, a member of the Atg8 family of proteins, along with the induction of autophagy, occurs in response to androgen deprivation in prostate cancer cells. This alteration promotes the survival and proliferation of these cancer cells [171]. In addition, transcription factor EB (TFEB) is considered the transcriptional regulator for autophagy and is involved in the control of lysosomal biogenesis. In the context of prostate cancer, AR has been found to stimulate the expression of TFEB, thereby promoting autophagy induction. Additionally, AR influences various upstream mediators of autophagy, including ATG4B, ATG4D, ULK1, and ULK2, which contribute to the progression of prostate cancer. Prior research has demonstrated that AR-mediated autophagy induction is essential for the proliferation and viability of prostate cancer cells. Cao et al. indicated that inhibiting the mTOR pathway resulted in the induction of autophagy and increased the sensitivity of cancer cells to radiotherapy [172]. Blocking apoptosis or inhibiting caspases has been shown to enhance the induction of autophagy. Lin et al. suggested that forkhead box protein M1 (FOXM1) is involved in prostate tumorigenesis and metastasis. FOXM1 plays a role in conferring resistance to docetaxel-mediated chemotherapy in castration-resistant prostate cancer by activating the AMPK/mTOR-mediated autophagy pathway [173]. Accumulating evidence has shown that targeting autophagy alone or in combination with chemotherapy has been effective at enhancing cell death and improving the efficacy of cancer therapies.

#### 3.4.2. Bladder Cancer

Bladder cancer is the fourth most lethal urological malignant tumor in men and the sixth most common malignancy in the USA [174]. Clinically, autophagy may be a potential therapeutic strategy for treating human bladder cancer patients. Zhu et al. provided evidence that Atg7 was significantly upregulated in invasive bladder cancer, and silencing Atg7 expression led to a significant reduction in bladder cancer invasion. This suggests that Atg7 plays a role in regulating the invasion of bladder cancer cells [175]. Therefore, inhibiting autophagy might enhance the cytotoxic effects of chemotherapy and radiotherapy [176,177] (Table 4).

Kang et al. [178] demonstrated that knockdown of Atg12 by small interfering RNA (siRNA) for inhibiting autophagosome formation could enhance the anticancer effect of epidermal growth factor receptor (EGFR) inhibitors, lapatinib or gefitinib, on bladder cell cells. In addition, inhibition of autophagy by Atg12-siRNA combined with EGFR inhibitors increased apoptotic cell death, as confirmed by flow cytometry analysis. The data suggest that autophagy acts as a protective mechanism in bladder cell cells. In addition, Lin et al. conducted a study in which they demonstrated that suppressing autophagy by silencing Atg7/Atg12 or Beclin1 using shRNA resulted in a synergistic enhancement of the anticancer efficacy of cisplatin in bladder cell lines 5637 and T24. Their findings indicate that the combination of cisplatin with autophagy inhibitors increases the susceptibility of bladder cells to cisplatin treatment, thereby amplifying the cytotoxic effects of the chemotherapy drug [179]. 

Bacillus Calmette-Guérin (BCG) is another effective medication employed in the treatment of non-muscle-invasive bladder cancer (NMIBC). As many as 35% of NMIBC patients receiving intravesical BCG instillation have BCG-induced cystitis [180]. Cystitis manifests with symptoms such as increased urinary frequency, urinary incontinence, dysuria, micturition pain, and even noticeable hematuria [180]. Buffen et al. [181] conducted both in vivo and in vitro studies on the treatment of bladder cancer with BCG, revealing a correlation between rs3759601 in ATG2B and the progression and recurrence of bladder cancer. Furthermore, their research suggests that BCG intravesical instillation therapy may serve as a pivotal event influencing epigenetic alterations related to innate immunity. Moreover, BCG has been found to increase the levels of cleaved-caspase-3, LC-3BII, and Atg3, suggesting the induction of both cell apoptosis and autophagy in gastric cancer cells [182].
ijms-24-14887-t004_Table 4Table 4Autophagy inhibitors and activators in bladder cancer.InhibitorsMechanism of ActionCombined DrugBC LineReferenceChloroquine Lysosomal lumen alkalizerCisplatin, radiotherapy, lapatinibEJ, T24, RT-112, 5637, J82[176,177,178,179]3-Methyladenine PI3K inhibitorCisplatin, Fangchinoline, lapatinibRT-112, T24, J82[177,178,179]shRNAKnockdown of Beclin1 and ATG7/ATG12Cisplatin5637, T24[179]siRNA Suppression of ATG12Lapatinib or gefitinibT24, J82[178]Activator



Ubenimex Akt agonistAkt5637, RT112[183]SalidrosideSuppressing PI3K and p-Akt Autophagy/PI3K/Akt T24[184]TetrandrineUpregulating p-AMPK and downregulating p-mTOR AMPK/mTORT24, 5637[185]Note: BC, bladder cell; shRNA, short hairpin RNA; siRNA, small interfering RNA; ATG, autophagy-related protein; AMPK, AMP-activated protein kinase; p-, phosphorylated.


#### 3.4.3. Renal Cancer

Renal cell carcinoma (RCC) is the most prevalent type of renal cancer. Among all cancer types, clear cell RCC (ccRCC) constitutes approximately 87% of RCC cases [186]. Currently, the primary treatment method for RCC is surgical tumor removal, often through procedures such as partial or radical nephrectomy, depending on the stage and extent of the cancer. In some cases, additional treatments, such as targeted therapies, immunotherapies, or radiation therapy, may be considered, especially for advanced or metastatic RCC. Autophagy plays a significant role in RCC and is also a potential therapeutic approach. Wang et al. discovered that LC3-II expression levels in RCC cell lines (786-O, 769-P, OS-RC-2, and ACHN cells) were lower compared to those in a control cell line (HK-2 cells) [187]. A high expression of Beclin 1 has been identified in the tissues and cells of RCC [188]. Beclin 1 plays a crucial role in inducing autophagy [189]. By binding to Beclin 1, Bcl-2 acts as an inhibitor of autophagy [190]. 

In RCC, the assessment of specific combinations of ATG1, ATG16L1, ATG5, LC3B, and p62, all of which gauge the basal level of autophagy, has demonstrated the ability to distinguish between normal tissue, clear cell RCC, and chromophobe RCC [191]. This suggests that the basal level of autophagy can serve as a potentially valuable parameter for discriminating among different types of RCC [191]. The dysregulation of autophagy and alterations in ATG expression can impact the development and progression of kidney cancer. The low expression of ATGs is predictive of a poor prognosis in RCC [192]. Wang et al. discovered that enhancing Atg7 resulted in reduced growth in OS-RC-2 cells, as confirmed by studies conducted in vitro using a xenograft animal model. Conversely, knocking down Atg7 increased cell growth. These findings strongly suggest that a defect in autophagy is a critical factor contributing to the growth of RCC cells [187].

Co-administering chloroquine (which affects the Bcl-2 family) with everolimus promotes cancer cells’ shift from autophagy to apoptosis, ultimately enhancing the therapeutic effectiveness of everolimus [190]. Silibinin induces early autophagy in RCC cells, leading to the inhibition of migration and invasion in vitro via AMPK/mTOR pathway activation, as evidenced by increased LC3-II expression and autophagolysosome vacuole formation [193]. To further substantiate the influence of licochalcone A (LCA) on autophagy, Hong et al. conducted a study involving pre-treatments using LY294002 and rapamycin on the PI3K/Akt/mTOR signaling pathway. The findings revealed that LCA inhibited the PI3K/Akt/mTOR signaling pathway, resulting in heightened autophagy in renal cancer cells [194].

#### 3.4.4. Testicular Cancer 

In testicular cancer, testicular germ cell tumors are the most common type and are predominantly found in males between the ages of 15 and 40 [195]. In testicular cancer, the expression levels of autophagy-related proteins BAG3 and p62 have been intensively examined in several studies. BAG3 is notably elevated in seminoma compared to both non-seminoma and normal testicular tissue. However, there is no significant difference in p62 expression detected between neoplastic and normal tissue or between seminoma and non-seminoma [196]. For the management of advanced testicular cancer, platinum-based neoadjuvant or adjuvant chemotherapy is essential. One contributing factor to the failure of chemotherapy is the stimulation of autophagy in cancer cells by chemotherapy drugs. This stimulation helps maintain mitochondrial function, reduces DNA damage, and supplies nutrients, such as amino acids and ATP to cancer cells. As a result, cancer cells experience an extended survival period, ultimately leading to the development of drug resistance [197]. 

In a study conducted by Zhu et al., inhibiting autophagy by silencing ATG5 and ATG7 enhanced the effectiveness of cisplatin in inhibiting cisplatin-resistant I-10 testicular cancer cells [198]. Furthermore, another study observed that, following the expression of connexin43, there was a decrease in the expression of the autophagy marker protein LC3-II, coupled with an increase in the expression of the autophagy substrate protein p62. These findings suggest that connexin43 protein may inhibit autophagy in cisplatin-resistant testicular cancer cells. [199]. It has also been observed that Pannexin1 inhibits autophagy in cisplatin-resistant testicular cancer cells by facilitating the release of ATP [200]. Overall, autophagy plays a role in testicular cancer, and manipulating or controlling autophagy could be a promising therapeutic approach for addressing this type of cancer.

#### 3.4.5. Penile Cancer

Penile cancer is a rare type of cancer that occurs on the foreskin, the glans, or on the skin of the penile shaft. In the United States, penile cancer makes up less than 1% of all cancers diagnosed in men. A possible cause of penile cancer is related to human papillomavirus (HPV) infection. HPV is a virus that passes through sex. Antibodies to HPV-16 have been found in many patients with penile cancer. The overexpression of HPV E5, E6, and E7 oncoproteins can alter the cellular proteins involved in cell proliferation, apoptosis, and immortalization. HPV infection can initially deactivate autophagy, but in the context of cancer, it may reactivate autophagy in advanced stages [201].

## 4. Future Research Avenues, Monitoring Methods, and Unresolved Questions

Autophagy is regulated through the posttranslational modification of ATG proteins, encompassing processes such as phosphorylation, ubiquitination, acetylation, O-GlcNAcylation, N6-methyladenosine modification, oxidation, and cleavage. These modifications serve as indicators that can be monitored to assess the status of the autophagic process [202,203,204,205,206,207,208]. There are many methods for monitoring autophagy, including transmission electron microscopy (TEM), Western blot, immunofluorescence, flow cytometry, northern blot, reverse transcriptase polymerase chain reaction (RT-PCR), luciferase release assay, transgenic mice, and gene knockdown, to identify and quantify the status of autophagic flux in different pathological conditions.

### 4.1. Transmission Electron Microscopy (TEM)

TEM, known for its high-resolution capabilities, has proven to be a dependable technique for monitoring autophagy. It has been utilized to identify and quantify autophagosomes. TEM investigations have revealed that autophagosomes are double-membrane structures encapsulating undigested cytoplasmic content, including organelles. Conversely, autolysosomes are characterized as single-membrane structures harboring cytoplasmic components in various stages of degradation [209]. Immuno-TEM combined with gold-labeling, utilizing antibodies, can provide insights into specific attributes of the autophagic process. This technique allows for the visualization of particular molecules or proteins within the cellular structures involved in autophagy, offering a more detailed understanding of the process’s intricacies [210,211]. LC3 immunogold labeling offers the capability to identify degradative organelles present in autophagy compartments. Nevertheless, the application of TEM presents challenges. It is a relatively time-consuming method that demands technical proficiency to ensure accurate sample handling throughout all stages of preparation, from fixation to sectioning and staining.

### 4.2. Western Blot and Fluorescence Microscopy for Atg Proteins

Mammalian homologs of Atg8-family proteins are categorized into two principal subfamilies: MAP1LC3/LC3 and GABARAP. The identification of both LC3-I and LC3-II variants is typically conducted through Western blot analysis and immunofluorescence, employing anti-LC3 antibodies. The expressed location of autophagy has been evaluated by LC3 II and the Bcl2-interacting protein Beclin-1 detected by immunofluorescence. Additionally, the assessment of autophagic flux, indicative of LC3-II turnover and variations in LC3 levels, can be inferred based on the ratio of LC3-II to Atg8–PE and monitored using Western blot techniques. Therefore, the measurement and quantification of Atg8-family proteins are essential to studying the formation of autophagosomes. This can be accomplished by using either anti-GFP antibodies to target GFP-ATG8 fusion proteins, or by employing anti-ATG8 antibodies for direct labeling [212]. Using Atg8-family proteins (Atg8/LC3-II/GABARAP-II) visualized fluctuations in autophagy, the protein detection and quantification have been shown by time-lapse fluorescence microscopy (dynamic assessment) of Atg8 puncta combined with automated microscopy. Furthermore, protein deacetylation is correlated with autophagy and can be assessed through quantitative immunofluorescence and Western blotting using antibodies that specifically recognize acetylated lysine residues [213]. Moreover, the co-localization of markers specific to mitochondria or endoplasmic reticulum with lysosomal proteins has been employed to monitor autophagolysosome formation. Additionally, immunofluorescence or flow cytometry analysis has led to the creation of two distinct fluorescent probes. These probes are used to detect alterations in the activity of two cytosolic proteins and to track autophagic flux within cultured cells.

### 4.3. Northern Blot or Reverse Transcriptase Polymerase Chain Reaction (RT-PCR) for Assessing the mRNA Levels

Elevated autophagy flux in human tissues has been observed to correlate with the tissue expression of ATG4B, ATG5, ATG8, Atg7, LC3A, and LC3B proteins [214,215]. Therefore, evaluating the mRNA levels of LC3 and other autophagy-related genes through methods, such as northern blot and RT-PCR, may offer corroborative information concerning autophagy induction.

### 4.4. Luciferase Release Assay and Fluorescence Microscopic Analyses

The luciferase release assay may be well-suited for the study of autophagy flux. For example, the assay can measure autophagy flux by monitoring the proteolytic and deconjugating activity of LC3 or ATG4B [216,217]. Under nutrient depletion, the lipid anchor (phosphatidylserine or PE) from lipidated LC3 (LC3-II) is cleaved by ATG4B and results in luciferase release to reduce the punctate pattern of cytosolic LC3 expression. LC3-II expression can be performed by immunoblotting and punctate localization of GFP-LC3, whereas LC3 degradation after autophagosomal-lysosomal fusion can be shown by flow cytometry combined with lysosomal inhibitor treatment.

### 4.5. Transgenic Mouse Expressing the Luciferase Release Reporter System with GFP-LC3 Localization

Transgenic mice that express GFP fused to Atg8 or LC3 systemically are commonly used to study mammalian autophagic flux in vivo. This approach provides a useful tool for tracking autophagosomes, as GFP serves as a marker protein [218] (Table 5). When GFP-Atg8 or GFP-LC3 is transported to the lysosome/vacuole, the Atg8 or LC3 component of the chimera is susceptible to degradation, while the GFP portion remains resistant to hydrolysis. LC3B or a protein labeled at the N-terminus with a fluorescent protein, such as GFP-LC3 or GFP-Atg8, has been employed to track autophagy. This monitoring has been conducted through immunofluorescence microscopy, observing an elevation in punctate LC3 or GFP-LC3 structures [219,220]. Moreover, Beclin 1-GFP puncta, observable through fluorescence microscopy or TEM, can serve as an additional marker indicating the induction of autophagy [221].

Cryosections of tissue fixed with 3.8% paraformaldehyde are prepared and subjected to co-localization studies involving GFP-LC3 direct fluorescence (green) and indirect immunostaining for RFP (red). To quantitatively assess GFP/RFP puncta, sections are immunostained for GFP using a secondary antibody tagged with red fluorescence. Colocalization with GFP fluorescence is then analyzed using confocal microscopy. The upregulation of Atg is quantified by combining fluorescence microscopy with automated microscopy techniques.

### 4.6. Knockdown of Autophagy Gene

MicroRNAs (miRNAs), RNA interference (RNAi), short hairpin RNA (shRNA), and RNA interference (RNAi) techniques can be employed to downregulate gene expression, offering direct evidence of the role of autophagic components. Therapeutic strategies based on siRNA involve introducing synthetic siRNA molecules into target cells to induce RNA interference (RNAi), thereby suppressing the expression of specific messenger RNA (mRNA) and achieving gene silencing. The therapeutic potential of siRNAs and miRNAs has been demonstrated in various diseases, including cancers [222,223,224] and infections [225]. Nevertheless, the effectiveness of gene knockdown can vary based on the cell type and the stability of the targeted protein.

## 5. Conclusions

Autophagy plays a dual role in managing urologic diseases; for instance, activating autophagy can alleviate symptoms of CYP-induced cystitis. Conversely, inhibiting autophagy can potentially hinder the progression of testicular cancer into the cisplatin-resistant stage. While there is evidence that autophagy influences these processes, the precise molecular mechanisms governing these interactions in urologic diseases have yet to be fully characterized. On the other hand, identifying potential biomarkers associated with autophagy is a crucial step towards a deeper understanding of its role in these diseases.

## Figures and Tables

**Figure 1 ijms-24-14887-f001:**
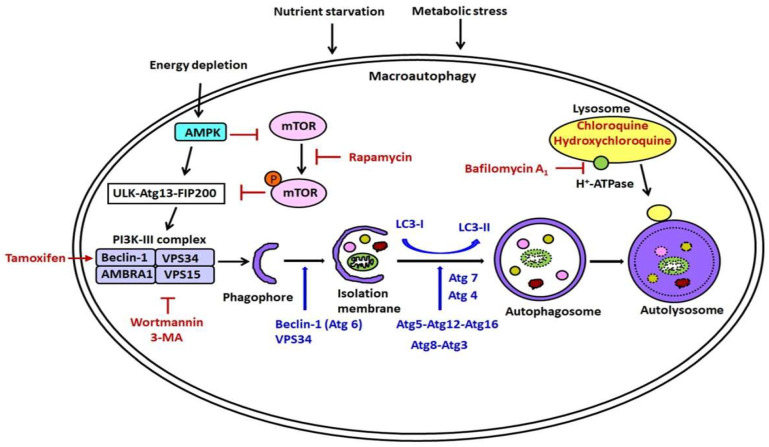
The molecular mechanism and pharmacological regulation for the macroautophagic pathway. The process of macroautophagy has three stages, described as follows: autophagy induction (initiation), autophagosome (phagophore; isolation membrane) formation, and autolysosome formation. (1) Autophagy induction: During instances of metabolic stress or nutrient scarcity, mTOR inhibition serves as a catalyst for autophagy initiation. This occurs through the assembly of the ULK complex, involving ULK, Atg13, and FIP200. Subsequently, the ULK complex phosphorylates AMBRA1, which triggers the activation of the PI3K complex, comprising VPS15, VPS34, Beclin-1, and AMBRA1. Additionally, AMP-activated protein kinase (AMPK), subject to AMP levels, exerts a negative regulatory influence on mTOR. AMPK also directly phosphorylates ULK, consequently functioning as a positive regulator of autophagy in response to energy depletion. (2) Autophagosome formation: The extension and enlargement of the phagophore membrane are governed by two systems of conjugation similar to ubiquitin: Atg12-Atg5-Atg16 and Atg8-Atg3. In the Atg12-Atg5-Atg16 system, Atg12 is initially activated through the ATP-dependent activity of the E1-like activating enzyme Atg7. Subsequently, Atg12 is transferred to the E2-like conjugating enzyme Atg10, forming the Atg12–Atg10 intermediate. Finally, conjugation occurs between the Atg5-Atg12 heterodimer and the Atg16 homodimer, leading to the creation of the Atg12-Atg5-Atg16 complex. The second ubiquitin-like system facilitates the attachment of phosphatidylethanolamine (PE) to Atg8/LC3. This process is sequentially managed by Atg4, Atg7, and Atg3. Consequently, nonlipidated LC3-I is transformed into the lipidated form, LC3-II. (3) Autolysosome formation: Autolysosome formation originates from the fusion of the autophagosome with the lysosome. Within autolysosomes, the enclosed contents are subjected to degradation by hydrolases, promoting recycling processes. Note: AMBRA1, activating molecule in Beclin 1-regulated autophagy protein 1; AMPK, adenosine 5′-monophosphate–activated protein kinase; Atg, autophagy-related gene; LC3, microtubule-associated protein 1 light chain 3; PE, phosphatidylethanolamine; PI3K, class III phosphotidylinositol-3-kinase; PI3P, phosphatidylinositol-3-phosphate; ULK, Unc-51–like kinase 1. The different colored circles in the autophagosome refer to various damaged cellular particles.

**Table 2 ijms-24-14887-t002:** Major autophagy related proteins/genes in mammals for autophagosome formation.

Autophagic Proteins	Gene	Yeast Homologue	Function in Autophagy Regulation	Reference
ATG1	*ULK1*	Atg1	A protein kinase that operates in the recruitment and release of other Atg proteins and is characterized by its serine/threonine activity. Initiator of autophagy: Atg1/ULK1 is a central component of autophagy.	[28,47,48,49]
ATG3	*ATG3*	Atg3	Atg3 is an ubiquitin-conjugating-like enzyme (E2) that conjugates Atg8-family proteins (Atg8/LC3) to PE after activation of the C-terminal residue by Atg7.	[50,51]
ATG4	*ATG4*	Atg4b	Atg8 cysteine cysteine protease. Atg 4 cleaves the C-terminus of Atg8/LC3 to expose a glycine residue, converts pro-LC3 (Atg8) to LC3-I, and forms autophagosomal LC3-II.	[28,47,48,49]
ATG5	*ATG5*	Atg5	Atg5 is covalently attached to Atg12 and binds Atg16. Atg 5 forms a complex with Atg12 and assists in autophagosomal elongation.	[52]
Beclin 1	*BECN1*	Atg6	Atg6 is a component of the PI3K complex. Activation of PI3K complex promotes autophagosomal membrane nucleation.	[28,47,48,49]
ATG7	*ATG7*	Atg7	An ubiquitin-activating (E1) enzyme. Atg 7 activates the conjugation of Atg8 proteins to PE, leading to the lipidation of Atg8-family proteins. Atg 7 acts as an E1 enzyme for Atg12 conjugation to Atg5 and Atg3, which is crucial for the autophagosome formation.	[53,54]
MAP1LC3B	*MAP1LC3B*	Atg8	Atg8/LC3 is an ubiquitin-like protein that is conjugated to PE and involved in cargo recruitment into phagophores and autophagosome biogenesis.	[28,47,48,49]
ATG9	*ATG9*	Atg9	Atg9 is a transmembrane protein that acts as a lipid carrier for expansion of the phagophore and assists in autophagosomal assembly.	[28,47,48,49]
ATG10	*ATG10*	Atg10	An ubiquitin-conjugating enzyme (E2) analog that conjugates Atg12 to Atg5	[55]
ATG12	*ATG12*	Atg12	An ubiquitin-like protein covalently modifies an internal lysine residue of Atg5 by binding to it via its C-terminal glycine. Atg12 forms a complex with Atg5 and assists in autophagosomal elongation.	[52,56]
ATG13	*ATG13*	Atg13	Supramolecular assembly of the Atg1 complex is initiated through the tethering of two Atg17 molecules by Atg13, which is essential for autophagy initiation.	[57]
ATG14	*ATG14*	Atg14	Atg 14 is an autophagy-specific subunit of the Beclin 1–class III phosphatidylinositol complex.	[58]
ATG16	*ATG16*	Atg16	Atg16 bound with Atg5 is associated with an isolation membrane in complex with Atg 5–Atg 12. Atg 16 assists in autophagosomal elongation.	[28,47,48,49]

Note: Atg, autophagy-related gene; ULK, Unc-51–like kinase 1; PI3K, class III phosphotidylinositol-3-kinase; PI3P, phosphatidylinositol-3-phosphate; MAP1LC3B, Microtubule-associated protein 1 light chain 3B; PE, phosphatidylethanolamine.

**Table 5 ijms-24-14887-t005:** Recommended detectors and methods for monitoring autophagy [49,216].

Detector of Autophagy	Methods and Assays
Atg8-family proteins LC3-II protein amount	Western blotting, mass spectrometry, confocal microscopy, flow cytometry, immunoblotting.
Autophagosome-lysosome colocalization	Fluorescence microscopy and quantitative fluorescence analysis.
Atg4B-dependent release of Gaussia luciferase (GLUC)	GLUC is a reporter enzyme. Luciferase release assay and fluorescence microscopic analyses.
Autophagosome quantification	Flow cytometry
Protein-proteim interaction	Bimolecular fluorescence complementation
GFP-Atg8-family protein	Fluorescence microscopy and flow cytometry to monitor vacuolar/lysosomal localization. Punctate GFP-Atg8-family protein or Atg18/WIPI, and live time-lapse fluorescence microscopy to monitor the dynamics of GFP-Atg8- family protein-positive structures.
Immunofluorescence for endogenous LC3 puncta GFP-LC3	Immunofluorescence can be used to identify autophagosomes in cells and transfect with a GFP-LC3 chimera.
Monitor autophagosome number, volume, and content/cargo.	Electron microscopy including quantitative electron microscopy, immuno-TEM.
mTOR, AMPK and Atg1/ULK1 kinase activity	Western blot, immunoprecipitation, or kinase assays.
GFP-Atg8 GFP-LC3	Assays can monitor selective types of autophagy by fluorescence microscopy and quantitative fluorescence analysis.
Tandem mRFP/mCherry -GFP fluorescence microscopy	Autophagic flux can be assessed by observing changes in fluorescence colors. Specifically, a decrease in green fluorescence (indicating phagophores and autophagosomes) and an increase in red fluorescence (indicating autolysosomes) can serve as indicators of ongoing autophagic activity.
Tissue fractionation	Centrifugation, Western blot, and electron microscopy for autophagic morphology.
Transcriptional and translational regulation	Northern blot, RT-PCR, or autophagy-dedicated microarray.
Turnover of autophagic compartments	Electron microscopy with morphometry/stereology at different time points.

## Data Availability

Not applicable.

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
