# Peer review of "The Molecular Mechanism and Therapeutic Application of Autophagy for Urological Disease"

_ijms, 2023, doi:10.3390/ijms241914887_

Round 1

Reviewer 1 Report

Dear authors,

Thank you for this creative review. Such works, as yours, are useful for science totally, not only for urology. I could suggest it for publication in its current form, but I have a minor argument for some terminology.

Actually, you document the term Partial Bladder Outlet Obstruction, which is not recommended by any global scientific committee (EAU, ICS). I could accept it as terminology, even in this form, if you had explained and described it in the manuscript. However, in the main text below this term, you are dealing - correctly - with the classical BOO. 

So, I suggest either to replace Partial BOO with BOO itself, or to give an analytic description about Partial BOO.

Author Response

Comments and Suggestions for Authors

Dear authors,

Thank you for this creative review. Such works, as yours, are useful for science totally, not only for urology. I could suggest it for publication in its current form, but I have a minor argument for some terminology.

  1. Actually, you document the term Partial Bladder Outlet Obstruction, which is not recommended by any global scientific committee (EAU, ICS). I could accept it as terminology, even in this form, if you had explained and described it in the manuscript. However, in the main text below this term, you are dealing - correctly - with the classical BOO.

So, I suggest either to replace Partial BOO with BOO itself, or to give an analytic description about Partial BOO.

Response: Thanks for your recommendation. We changed all the term “PBOO” to “BOO”. Please refer to pages 1, 2, 10 and 13. All the changes made in the manuscript are marked in red font on a yellow background

Reviewer 2 Report

This medical-scientific article provides an overview of the molecular relationship between autophagy and apoptosis, emphasizing their roles in various physiological and pathological conditions. The article also discusses the pharmacological regulation of autophagy and the methods used to study and monitor autophagy, including transmission electron microscopy (TEM), western blot, fluorescence microscopy, and gene knockdown techniques. Furthermore, it delves into the role of autophagy in specific diseases, such as bladder cancer. Certainly, here are the major revisions and improvements needed for the provided text:

1. Introduction

1.1. The Molecular Relationship Between Autophagy and Apoptosis

  • The introduction should begin with a more concise and engaging statement that highlights the importance of the topic.
  • Define and explain key terms like apoptosis, autophagy, and mitoptosis before diving into the pathways.
  • The first paragraph contains an overload of technical information. Simplify the language and avoid using too many acronyms.
  • Ensure that all references are cited properly for scientific credibility.
  • Provide context for why understanding the relationship between autophagy and apoptosis is important and the potential implications for diseases.

1.2. Molecular Mechanisms of Autophagy

  • Break down the sections into subsections for better organization.
  • Improve the clarity and coherence of the text by using clear transitions between ideas.
  • Consider using bullet points or numbered lists to explain the three types of autophagy for better readability.
  • Explain the significance of autophagic flux more explicitly.
  • Simplify and clarify complex processes such as autophagy induction, autophagosome formation, and autolysosome formation. Use diagrams or figures if necessary.
  • Be more concise in descriptions, and avoid excessive technical jargon without proper explanation.
  • Make sure to mention the clinical relevance of autophagy in each section and how it relates to the molecular mechanisms.

2. Conclusion

  • Summarize the key takeaways from the article in a clear and concise manner.
  • Emphasize the significance of understanding autophagy and apoptosis in the context of human diseases.
  • Suggest potential areas for future research or practical applications.

General Improvements

  • The text needs to be more reader-friendly. Simplify complex scientific concepts for a broader audience.
  • Use clear and concise language throughout the article.
  • Consider including more visual aids, such as figures or diagrams, to help illustrate complex processes.
  • Ensure consistent formatting and citation style throughout the article.
  • Ref. 91 should be formatted. 
  • Please include and discuss the following papers: PMID: 37446024; PMID: 36294423

This major revision should aim to make the content more accessible to a wider audience while retaining its scientific accuracy and credibility.

Author Response

Comments and Suggestions for Authors

This medical-scientific article provides an overview of the molecular relationship between autophagy and apoptosis, emphasizing their roles in various physiological and pathological conditions. The article also discusses the pharmacological regulation of autophagy and the methods used to study and monitor autophagy, including transmission electron microscopy (TEM), western blot, fluorescence microscopy, and gene knockdown techniques. Furthermore, it delves into the role of autophagy in specific diseases, such as bladder cancer. Certainly, here are the major revisions and improvements needed for the provided text:

  1. Introduction

1.1. The Molecular Relationship Between Autophagy and Apoptosis

The introduction should begin with a more concise and engaging statement that highlights the importance of the topic.

Define and explain key terms like apoptosis, autophagy, and mitoptosis before diving into the pathways.

The first paragraph contains an overload of technical information. Simplify the language and avoid using too many acronyms.

Ensure that all references are cited properly for scientific credibility.

Provide context for why understanding the relationship between autophagy and apoptosis is important and the potential implications for diseases.

Response: Thank you for your suggestion. We provided a updated paragraph briefly summarize the aim of our study, the topic and the role of our paper in the current literature. Please refer to pages 2.

  1. 1.2. Molecular Mechanisms of Autophagy

Break down the sections into subsections for better organization.

Improve the clarity and coherence of the text by using clear transitions between ideas.

Consider using bullet points or numbered lists to explain the three types of autophagy for better readability.

Explain the significance of autophagic flux more explicitly.

Simplify and clarify complex processes such as autophagy induction, autophagosome formation, and autolysosome formation. Use diagrams or figures if necessary.

Be more concise in descriptions, and avoid excessive technical jargon without proper explanation.

Make sure to mention the clinical relevance of autophagy in each section and how it relates to the molecular mechanisms.

Response: Thank you for your suggestion. Three types of autophagy have been explained using bullet points to enhance readability (Please refer to pages 4, line 141 to 151). Furthermore, the manuscript has also utilized graphics to simplify and elucidate complex processes such as autophagy induction, autophagosome formation, and autolysosome formation (Please refer to Figure 1 on pages 6).

  1. Conclusion

Summarize the key takeaways from the article in a clear and concise manner.

Emphasize the significance of understanding autophagy and apoptosis in the context of human diseases.

Suggest potential areas for future research or practical applications.

Response: As suggested by the reviewer, we have revised the Conclusion section (Please refer to pages 20).

  1. General Improvements

The text needs to be more reader-friendly. Simplify complex scientific concepts for a broader audience.

Use clear and concise language throughout the article.

Consider including more visual aids, such as figures or diagrams, to help illustrate complex processes.

Ensure consistent formatting and citation style throughout the article.

Ref. 91 should be formatted.

Please include and discuss the following papers: PMID: 37446024; PMID: 36294423

This major revision should aim to make the content more accessible to a wider audience while retaining its scientific accuracy and credibility.

Response: Thanks for your professional recommendation. We have reconstructed Chapter 1, and imported Chapter 2 and Chapter 3 and have added more therapeutic strategies involving autophagy, therapeutic efficacy and limitation for various urological diseases. Additionally, this article follows the reference style established in the published papers of IJMS, which mandates listing all authors' names uniformly. In addition, the following papers were also included (PMID: 37446024; PMID: 36294423) (Please refer to Ref. 60 and 61).

Reviewer 3 Report

General comment

The manuscript entitled “The molecular mechanism and therapeutic application of autophagy for urological disease” by Chueh et al., aims to extensively examine the molecular mechanisms of autophagy in urological diseases in order to evaluate potential pharmacological and therapeutic involvement in clinical practice. The manuscript deals with an interesting topic albeit, as it is written, it seems a wall-text, potentially confusing the reader in some points. To this regard, few stylistic correction are suggested in addition to improve the work adding few missing data.

INTRODUCTION

The introduction slowly turns into a paragraph without providing, indeed, a proper introduction. I would revise the first paragraph, adding a proper introduction in which you briefly summarize the aim of your study, the topic and the role of your paper in the current literature.

Table 1 could avoid the bullet list aspect by removing the 1. 2. 3. Etc.

Paragraphs 1.2 and 1.1 have a too similar title.

Considering the content of paragraph 1.2, the first table should be placed at the end of this paragraph.

Line 136 could represent the beginning of a new subparagraph or, at least, provide a blank line to improve the readability.

Similar issues for line 146 and 156

POTENTIAL THERAPEUTIC APPLICATION OF AUTOPHAGY IN UROLOGICAL DISEASES

Considering that this represents the main paragraph of the work, the previous ones could be synthesized.

Regarding the different types of cystitis, I would add also BCG-induced cystitis.

In urological cancer you should also report, at least, renal and testicular cancer. See also for further data 10.3390/medicina59040724

Another cancer that should be also considered is penile cancer.

Minor editing of English language required

Author Response

  1. The introduction slowly turns into a paragraph without providing, indeed, a proper introduction. I would revise the first paragraph, adding a proper introduction in which you briefly summarize the aim of your study, the topic and the role of your paper in the current literature.

Response: Thank you for your suggestion. We provided a updated paragraph briefly summarize the aim of our study, the topic and the role of our paper in the current literature. Please refer to pages 2.

  1. Table 1 could avoid the bullet list aspect by removing the 1. 2. 3. Etc.

Response: Thank you for your suggestion. The bullet list was removed. Please refer to Table 1 on pages 3 and 4, Table 2 on pages 7 and 8, Table 3 on pages 9 and 10, and Table 5 on page 20.

  1. Paragraphs 1.2 and 1.1 have a too similar title.

Response: Thanks for your professional recommendation. As suggested by the reviewer, the title for Paragraphs 1.1 was changed to “2.1. Comparison between autophagy and apoptosis”. Please refer to page 2.

Location

Original content

Revised content

Page 2

1.1. The molecular relationship between autophagy and apoptosis

2.1. Comparison between autophagy and apoptosis

  1. Considering the content of paragraph 1.2, the first table should be placed at the end of this paragraph.

Response: As suggested by the reviewer, we have moved Table 1 to the end of this paragraph. Please refer to pages 3 and 4.

  1. Line 136 could represent the beginning of a new subparagraph or, at least, provide a blank line to improve the readability. Similar issues for line 146 and 156.

Response: Since these contents are coherent, we decided to maintain the original article format.

  1. POTENTIAL THERAPEUTIC APPLICATION OF AUTOPHAGY IN UROLOGICAL DISEASES

Considering that this represents the main paragraph of the work, the previous ones could be synthesized.

Response: Thanks for your professional recommendation. As suggested by the reviewer, we have revised and restructured the layout of the article on Paragraph 3 as “Potential therapeutic application of autophagy in urological diseases” (Please refer to page 10 to 17).

  1. Regarding the different types of cystitis, I would add also BCG-induced cystitis.

Response: Thank you for your professional suggestion. However, there is currently no direct evidence linking autophagy and BCG-induced cystitis. Nevertheless, autophagy does play a significant role in the treatment of bladder cancer with BCG. We provided additional details in the bladder cancer paragraph. Please refer to page 15 and 16.

  1. In urological cancer you should also report, at least, renal and testicular cancer. See also for further data 10.3390/medicina59040724

Another cancer that should be also considered is penile cancer.

Response: As suggested by the reviewer, we have added renal, testicular and penile cancer paragraphs (Please refer to pages 16 and 17).

Reviewer 4 Report

The Chapter 1-3 of this review suffers from a lack of depth and coherence, largely due to its ambitious scope in covering a wide array of autophagy-related phenomena. This approach risks losing the reader's engagement. To enhance the review, the authors could concentrate more on Chapter 4, titled "Potential Therapeutic Application of Autophagy in Urological Diseases." While this chapter offers a unique angle within the review, it falls short in adequately explaining the fundamental biological underpinnings and mechanisms of autophagy in relation to various diseases. A more effective strategy might be to eliminate the first half of the review and incorporate its content into Chapter 4, thereby providing a foundational understanding of autophagy's role in each disease. Furthermore, the review lacks comprehensive details on specific therapeutic strategies involving autophagy, as well as insights into their efficacy and limitations. The inclusion of future research avenues and unresolved questions would also enrich the review.

Author Response

  1. The Chapter 1-3 of this review suffers from a lack of depth and coherence, largely due to its ambitious scope in covering a wide array of autophagy-related phenomena. This approach risks losing the reader's engagement. To enhance the review, the authors could concentrate more on Chapter 4, titled "Potential Therapeutic Application of Autophagy in Urological Diseases." While this chapter offers a unique angle within the review, it falls short in adequately explaining the fundamental biological underpinnings and mechanisms of autophagy in relation to various diseases. A more effective strategy might be to eliminate the first half of the review and incorporate its content into Chapter 4, thereby providing a foundational understanding of autophagy's role in each disease.

Response: Thank you for your suggestion. We have reconstructed Chapter 1, and imported Chapter 2 and Chapter 3 into the Chapter “Future research avenues, monitoring methods and unresolved questions” (Please refer to pages 17 to 20).

  1. Furthermore, the review lacks comprehensive details on specific therapeutic strategies involving autophagy, as well as insights into their efficacy and limitations.

Response: Thanks for your professional recommendation. As suggested by the reviewer, we have added more therapeutic strategies involving autophagy, therapeutic efficacy and limitation for various urological diseases (Please refer to pages 8 to 18). We tried to integrate a large number of literatures, but the precise role of autophagy and therapeutic application for various urological diseases remains incompletely understood.

  1. The inclusion of future research avenues and unresolved questions would also enrich the review.

Response: As suggested by the reviewer, we have added the paragraph “Future research avenues, monitoring methods and unresolved questions.” Please refer to page 17 to 20.

Round 2

Reviewer 2 Report

the manuscript can now be considered for publication

Reviewer 3 Report

The authors improved the manuscript accordingo to suggestions.

However, properly revise this point

"In urological cancer you should also report, at least, renal and testicular cancer. See also for further data 10.3390/medicina59040724"

none

Reviewer 4 Report

I agree that this article is accepted because the authors have sufficiently addressed the points raised by my peer review.